# Regularization, Bayesian Inference, and Machine Learning Methods for Inverse Problems [note 1]

**DOI:** 10.3390/e23121673

**Published:** 2021-12-13

**Authors:** Ali Mohammad-Djafari

**Affiliations:** 1Laboratoire des Signaux et Système, CNRS, CentraleSupélec-University Paris Saclay, 91192 Gif-sur-Yvette, France; djafari@lss.supelec.fr; 2International Science Consulting and Training (ISCT), 91440 Bures-sur-Yvette, France; 3Scientific Leader of Shanfeng Company, Shaoxing 312352, China

**Keywords:** inverse problems, regularization, Bayesian inference, machine learning, artificial intelligence, Gauss–Markov–Potts, Variational Bayesian Approach (VBA), physics-informed ML

## Abstract

Classical methods for inverse problems are mainly based on regularization theory, in particular those, that are based on optimization of a criterion with two parts: a data-model matching and a regularization term. Different choices for these two terms and a great number of optimization algorithms have been proposed. When these two terms are distance or divergence measures, they can have a Bayesian Maximum A Posteriori (MAP) interpretation where these two terms correspond to the likelihood and prior-probability models, respectively. The Bayesian approach gives more flexibility in choosing these terms and, in particular, the prior term via hierarchical models and hidden variables. However, the Bayesian computations can become very heavy computationally. The machine learning (ML) methods such as classification, clustering, segmentation, and regression, based on neural networks (NN) and particularly convolutional NN, deep NN, physics-informed neural networks, etc. can become helpful to obtain approximate practical solutions to inverse problems. In this tutorial article, particular examples of image denoising, image restoration, and computed-tomography (CT) image reconstruction will illustrate this cooperation between ML and inversion.

## 1. Introduction

Inverse problems arise in almost any scientific and engineering application. In fact, they arise whenever we want to infer a quantity that is not directly measured. Noting the unknown quantity *f* and the measurement data *g*, we may have a mathematical relation between them: g=H(f) where *f* can be a 1D function (signal), a 2D function (image), a 3D function, or more (e.g., video, hyperspectral images, etc.). H is a mathematical model, called a forward operator, and *g* can also be a 1D, 2D, 3D, or more function. In practice, we may only have discrete values of it available, and, for this reason, the inverse problem that is inferring *f* from this limited data is an ill-posed problem. When discretized, we may write the relations between them as g=H(f)+ϵ where *g* contains all the data, *f* contains all the discretized representations of the unknown quantity, and *H* is a multidimensional operator connecting them. Finally, ϵ represents all the errors of discretization and measurement uncertainties.

Handling inverse problems, even in the discretized version linear model g=Hf+ϵ is not easy, at least for two reasons: one is the ill-conditioning of the matrix *H* and its great dimensions; the second is accounting for the errors.

Classical methods for inverse problems are mainly based on regularization theory, particularly those that are based on optimization of a criterion with two parts: a data–model-matching part $\Delta_{1}\left(g,Hf\right)$ and a regularization term $\Delta_{2}\left(f,f_{0}\right)$ with a balancing term between them: $J\left(f\right)=\Delta_{1}\left(g,Hf\right)+\lambda \Delta_{2}\left(f,f_{0}\right)$ where $\Delta_{1}$ and $\Delta_{2}$ are two distances (L2, L1, etc.) or divergence measure such as Kullback–Leibler (KL) or any other divergence, when, for example, *f* and $f_{0}$ are densities. $f_{0}$ can be equal to zero or any other prior default solution. Different choices for these two terms and a great number of optimization algorithms have been proposed with success in very diversified domains and applications [1,2,3,4,5].

Bayesian-inference-based methods also had great success for handling inverse problems, in particular, when the data are noisy, uncertain, some are missing and with some outliers, and where there is a need to account and to quantify uncertainties. In fact, the two terms of the regularization methods can have a Bayesian Maximum A Posteriori (MAP) interpretation where these two terms correspond to the likelihood and prior models, respectively. Indeed, the Bayesian approach gives more flexibility in choosing these terms and particularly the prior term via hierarchical models and hidden variables [6,7,8,9] However, the Bayesian computations can become very heavy computationally. The machine-learning (ML) methods, such as classification, clustering, segmentation, and regression, based on neural networks (NN), such as convolutional NN and deep NN, physics-informed neural networks, etc., can become helpful to obtain approximate but good-quality and practical solutions to inverse problems [10,11,12,13].

However, even if in many domains of machine learning such as classification and clustering these methods have shown success, their use in real scientific problems is limited. The main reasons are twofold. First, the users of these tools cannot explain the reasons why they are successful and why they are not. The second is that, in general, these tools cannot quantify the remaining uncertainties.

Model-based and Bayesian-inference approaches have been very successful in linear inverse problems. However, adjusting the hyperparameters is complex, and the cost of the computation is high. The convolutional-neural-networks (CNN) and deep-learning (DL) tools can be useful for pushing these limits further. On the other side, the model-based methods can be helpful for the selection of the structure of CNN and DL, which are crucial in ML success. In this tutorial article, first an overview and a survey of the aforementioned methods are presented, and the possible interactions between them are explored [14,15].

The rest of the article is organized as follows: First, a survey of inverse-problem examples, analytical inversion methods, generalized inversion and regularization methods, and finally the Bayesian inference methods is presented. Then, a discussion on the process and final objectives of imaging systems, for example, in health survey systems, going from the data acquisition to image reconstruction, its segmentation, its feature extraction, and finally its interpretation and usage is presented to prepare the more advanced part of this tutorial—for example, the Bayesian joint reconstruction and segmentation using Gauss–Markov–Potts prior modeling [16,17,18,19]. In the third part, first an introduction to machine-learning (ML) tools and processes and basic notions and notations on neural networks (NN) are given. The last part is related to the relations between all these methods via forward modeling, identification, learning, and inversion. These relations are shown via a few simple examples, and then we discuss the fully learned and physics-informed partially learned ML methods for inverse problems.

After mentioning some successful case studies in which the ML tools have been successful [20,21,22,23,24,25,26,27,28,29,30,31,32,33,34,35], we arrive at the main conclusions of this article and the future of the possible interactions between model-based and machine-learning tools. We conclude by mentioning the open problems and challenges in both the classical, model-based and the ML tool.

## 2. Inverse Problems Example

Inverse problems arise almost everywhere in science and engineering— everywhere we want to infer an unknown quantity *f* that is not accessible (observable) directly. We only have access to another observable quantity *g* that is related to it via a linear or a non-linear relation ***H*** [36,37,38].

As you can see, I am going to use a color code: red for unknown quantities and blue for observed or assumed, known quantities. The forward operator linking the two quantities is noted ***H***. In general, the forward operator is well-posed, but the inverse problem is ill-posed. This means that either the classical inverse operator does not exist (existence), or we can define many generalized inverse operators, so many solutions to the problem can be defined (uniqueness), or even if we can define an inverse operator, it may be unstable (stability) [39].

Let us mention a few examples of common inverse problems here.

### 2.1. Image Restoration

Any photographic system (camera, microscope, or telescope) has a limited field of view and a limited resolution. If we note by f(x,y) the original image and by the g(x,y) the observed image and if we assume a linear and space-invariant operator between them, then the forward relation can be written as a convolution operator:(1)$$g\left(x^{\prime },y^{\prime }\right)=\int f\left(x,y\right)h\left(x^{\prime }-x,y^{\prime }-y\right)\;dx\;dy\;,$$
where h(x,y) represents the point spread function (psf) of the imaging system.

Many examples can be given [40,41]. In Figure 1, two synthetic examples are shown.

### 2.2. X-ray Computed Tomography

In X-ray computed tomography (CT), the relation between the data and the object can be modeled via the radon transform:(2)g(r,ϕ)=∫∫f(x,y)δ(r−xcosϕ−ysinϕ)dxdy,
where δ is the Dirac function; thus, g(r,ϕ) represents the line integrals over the lines of angles ϕ of the object function f(x,y). Forward operation is called *projection*, and the inversion process is called *image reconstruction*. In Figure 2, one synthetic example is shown.

### 2.3. Acoustical Imaging

Acoustic source localization in acoustical imaging can also be considered as an inverse problem, where the positions and intensities of acoustical sources have to be estimated from the signal received by the microphone arrays. If we represent the distribution of the sources by
$$f\left(x,y\right)=\sum_{n}s_{n}\delta \left(x-x_{n},y-y_{n}\right)\;,$$
each microphone receives the sum of the delayed sources’ sounds [42,43,44,45]:$$g_{m}\left(t\right)=\sum_{n}s_{n}\left(t-\tau_{mn}\right)\;,$$
where $\tau_{mn}$ is the delay of transmission from the source position *n* to the microphone position *m*. This delay is a function of the speed of the sound and the distance between the source position $\left(x_{n},y_{n}\right)$ and the microphone position $\left(x_{m},y_{m}\right)$.

In Figure 3, one synthetic example is shown to explain the main idea.

### 2.4. Microwave Imaging for Breast-Cancer Detection

In microwave imaging, the body is illuminated by microwaves. As the electrical properties (conductivity and permeability) of the healthy and tumor tissues are different, their corresponding induced sources are different. These differences can be measured via the electrodes outside of the breast. The inverse problem, in this case, consists in estimating these induced sources or even directly the distribution of the conductivity and permeability inside the breast. Looking at such images, the tumor area can be visualized [46,47].

The forward problem here is described by two linked equations: $$\left\{\begin{array}{c} g\left(r_{i}\right)=\int \;\int_{D}G_{m}\left(r_{i},r^{\prime }\right)\phi \left(r^{\prime }\right)\;f\left(r^{\prime }\right)\;dr^{\prime },\;r_{i}\in S\;;\\ \phi \left(r\right)=\phi_{0}\left(r\right)+\int \;\int_{D}G_{o}\left(r,r^{\prime }\right)\phi \left(r^{\prime }\right)\;f\left(r^{\prime }\right)\;dr^{\prime },\;;\;r\in D \end{array}\right.$$
where f(r) represents the variation of the conductivity, ϕ(r) the induced field due to the incident field $\phi_{0}\left(r\right)$, and $G_{m}\left(r_{i},r^{\prime }\right)$ and $G_{o}\left(r,r^{\prime }\right)$ are the Green functions.

The first one relates the measured diffracted field on the sensors g(r) as a function of the induced currents J(r)=ϕ(r)f(r) inside the brain due to the external field via the Green functions, and the second relates the total field as the sum of the incident and the induced field. So, the forward problem is nonlinear, as ϕ(r) appears in both sides of the equation. However, it can be approximated by a linear relation if we assume that the induced field inside is very small compared to the incident field: ($\phi \left(r^{\prime }\right)≃\phi_{0}\left(r^{\prime }\right)$). This is called Born approximation: $$g\left(r_{i}\right)=\int \;\int_{D}G_{m}\left(r_{i},r^{\prime }\right)\phi_{0}\left(r^{\prime }\right)\;f\left(r^{\prime }\right)\;dr^{\prime },\;r_{i}\in S\;.$$

Both the bi-linear relations and the linear Born approximations are used in microwave imaging. The first one is more common in industrial non-destructive testing (NDT) and the second for biological and medical applications.

### 2.5. Brain Imaging

In brain imaging, the electrical activity of the neurons inside the brain brain are propagated and can be measured at the surface of the sculpt via the electrodes fixed on it. These signals are called electroencephalography (EEG). It is also possible to measure the magnetic field created by this activity. This time, the signals are called (MEG). In both cases, the inversion process consists in estimating the distribution of the brain activity from the measured signals. If the brain electrical activity can be modeled as the electrical mono- or dipoles distributed over the surface of the brain, then the simplified forward model can be almost similar to acoustical sources localization of the previous example. Here, the distribution of the sources are in the 3D space of the brain, and the EEG electrodes are positioned on the sculpt. The signal received by each EEG electrode can be compared to the signals received by the microphones in the previous example. There are a great number of forward models, analysis, and inversion methods that have been proposed for this application. A very good toolbox is the EEGLAB, which can be searched on the internet and easily obtained.

### 2.6. Other Applications

Many other imaging systems to see inside the human body or inside any industrial object in non-destructive testing (NDT) applications exist. Here, a few of them are illustrated. We can just mention a few more: magnetic-resonance imaging (MRI), ultrasound imaging such as echography, positron emission tomography (PET), single-emission computed tomography (SPECT), electrical impedance tomography, and eddy current tomography [48].

## 3. Classification of Inverse Problems’ Methods

Inverse problems’ methods can be classified in the following categories:Analytical inversion methods.Generalized inversion approach.Regularization methods.Bayesian inference methods.

In the first category, the main idea is to recognize the forward operators as one of the well-known mathematical invertible operator and thus to use the appropriate inversion operator. Typical examples are Fourier transform (FT) and radon transform (RT). In the second category, the notion of generalized inversion is used. The corresponding methods are either based on singular value decomposition (SVD) or the iterative projection based algorithms. The regularization methods are mainly based on the optimization of a criterion, often made in two parts: data–model adequacy and the regularization with a regularization parameter. Finally, the Bayesian inference approach, which I consider to be the most general and complete, has all the necessary tools to go beyond the regularization methods.

## 4. Analytical Methods

Figure 4 shows the main idea behind the analytical methods via two classical cases of image deconvolution and X-ray image reconstruction. In the first case, as the forward model is a 2D convolution: g(x,y)=h(x,y)∗f(x,y) or equivalently a 2D Fourier-transform (FT) filtering: G(u,v)=H(u,v)F(u,v); the operation consists in going to the Fourier domain, doing inverse filtering, and coming back. However, the inverse filtering $\frac{1}{H(u,v)}$ must be regularized either by limiting the band width or by applying an appropriate window mask before doing the inverse Fourier transform (IFT).

In the second case, the forward model is the radon transform (RT). Using the relation between FT and RT (Fourier slice theorem), the analytical inversion process becomes:1.For each angle ϕ, compute the 1D FT of $g_{\phi }\left(r\right)=g\left(r,\phi \right)$,2.Relate it to the 2D FT of f(x,y) via the Fourier slice theorem and interpolate to obtain the full 2D FT of f(x,y); and3.Compute 2D IFT to obtain f(x,y)

For more details, refer to [49,50].

The main difficulty with the analytical method is that the forward relations are given for continuous functions or densities. They can give satisfactory results if the inversion process is regularized and if the data are dense, complete, and without any measurement or discretization errors. In practical situations, rarely are all these conditions satisfied.

## 5. Generalized Inversion Approach

In this approach, the main idea is based on the fact that the forward operator is in general a singular one. This means there are many possible solutions to the inverse problem. In this approach, there are mainly two categories of methods. The first are based on singular-values decomposition (SVD). The second is based on optimization of a criterion such as the least squares (LS). In both, the main idea is to define a set of possible solutions, called generalized inverse solutions: (3)$$\left\{f^{†}:Hf^{†}=g\right\}$$
or pseudo solutions: (4)$$\{f^{†}:∥Hf^{†}-g{∥}^{2}<\epsilon \}\;.$$

Then, between those possible solutions, one tries to define a criterion, such as the minimum norm, to choose a solution. For the linear inverse problems, the corresponding solutions are given by
(5)$$f^{†}={\left[H^{t}H\right]}^{-1}H^{t}g=H^{†}g\;\;\;\;\mathrm{or}\;\;\;\;f^{†}=H^{t}{\left[HH^{t}\right]}^{-1}g=H^{†}g\;.$$

In great dimensional problems, even if we have these analytical expressions, in practice, the solutions are computed by using iterative optimization algorithms, for example, to optimize the LS criterion $J\left(f\right)={∥Hf-g∥}^{2}$ by a gradient-based algorithm: (6)$$f^{\left(k+1\right)}=f^{\left(k\right)}+\alpha H^{t}\left(g-Hf^{\left(k\right)}\right)\;,$$
with a stopping criteria or just after some fixed number of iterations. We will see in the next sections how this can lead to a deep-learning NN structure.

## 6. Model-Based and Regularization Approach

The model-based methods are related to the notions of the forward-model and the inverse-problems approach. Figure 5 shows the main idea:

Given the forward model H and the source *f*, the prediction of the data *g* can be done, either in a deterministic way: g=H(f) or via a probabilistic model: p(g|f,H) as we will see in the next section. In the same way, given the forward model H and the data *g*, the estimation of the unknown source *f* can be done either via a deterministic method or a probabilistic one. One of the deterministic methods is the generalized inversion: $f=H^{†}\left(g\right)$. A more general method is the regularization:(7)$$\hat{f}=\arg\min_{f}\left\{J(f)\right\}\;with\;J\left(f\right)={∥g-H\left(f\right)∥}^{2}+\lambda R\left(f\right)\;.$$

As we will see later, the only probabilistic method that can be efficiently used for the inverse problems is the Bayesian approach.

### Regularization Methods

Let consider the discretized linear inverse problem: g=Hf+ϵ, and the regularization criterion
(8)$$J\left(f\right)=\frac{1}{2}{∥g-Hf∥}_{2}^{2}+\lambda R\left(f\right)\;.$$

The first main issue in such a regularization method is the choice of the regularizer. The most-common examples are: (9)$$R\left(f\right)=\left\{{∥f∥}_{2}^{2}{,\;∥f∥}_{\beta }^{\beta }{,\;∥Df∥}_{2}^{2},\;{∥Df∥}_{\beta }^{\beta },\;\sum_{j}\phi \left({\left[Df\right]}_{j}\right)\right\},\;1\leq \beta \leq 2\;,$$
where ***D*** is a linear operator, generally a first-order derivation, a gradient, or a second-order derivation. The function ϕ has to be convex to ensure the uniqueness of the solution. Many such functions have been proposed, but some non-convex ones have also been proposed, which then need global optimization techniques.

The second main issue in regularization is the choice of an appropriate optimization algorithm. This mainly depends on the type of the criterion, and we have:R(f) quadratic: gradient-based and conjugate gradient algorithms are appropriate.R(f) non-quadratic but convex and differentiable: here too the gradient-based and the conjugate gradient (CG) methods can be used, but there are also a great number of convex criterion optimization algorithms.R(f) convex but non-differentiable: here, the notion of a sub-gradient is used.

Specific cases are:L2 or quadratic: $J\left(f\right)=\frac{1}{2}{∥g-Hf∥}_{2}^{2}+\lambda {∥Df∥}_{2}^{2}$;In this case we have an analytic solution: $\hat{f}={\left(H^{t}H+\lambda D^{\prime }D\right)}^{-1}H^{t}g$. However, in practice, this analytic solution is not usable in high-dimensional problems. In general, as the gradient $\nabla J\left(f\right)=-H^{t}\left(g-Hf\right)+2\lambda D^{\prime }Df$ can be evaluated analytically, gradient-based algorithms are used.L1 (TV): convex but not differentiable at zero: $J\left(f\right)=\frac{1}{2}{∥g-Hf∥}_{2}^{2}+\lambda {∥Df∥}_{1}$;The algorithms in this case use the notions of the Fenchel conjugate, the dual problem, the sub gradient, and the proximal operator [11,51,52,53]Variable splitting and augmented Lagrangian
(10)$$\left(f,\hat{z}\right)=\arg\min_{f,z}\left\{\frac{1}{2}{∥g-Hf∥}_{2}^{2}+{\lambda ∥z∥}_{1}+q{∥z∥}_{2}^{2}\right\}\;\;s.t.\;\;z=Df\;.$$A great number of optimization algorithms have been proposed: ADMM, ISTA, FISTA, etc. [1,5,54].

The main limitations of deterministic regularization methods are:A limited choice of the regularization term. Mainly, we have: (a) smoothness (Tikhonov) and (b) sparsity, piecewise continuous (total variation).Determination of the regularization parameter. Even if there are some classical methods such as the L-curve and cross validation, there are still controversial discussions about this.Quantification of the uncertainties: this is the main limitation of the deterministic methods, particularly in medical and biological applications where this point is important.

The best possible solution to push further all these limits is the Bayesian approach, which has: (a) many possibilities to choose prior models, (b) the possibility of the estimation of the hyperparameters, and, most important, (c) an accounting for the uncertainties.

## 7. Bayesian-Inference Methods

### 7.1. Basic Idea

The simple case of the Bayes rule is:(11)$$p\left(f|g,H\right)=\frac{p(g|f,H)\;\;p(f|H)}{p(g|H)}\;\mathrm{where}\;p\left(g|H\right)=\int \;\int p\left(g|f,H\right)\;\;p\left(f|H\right)\;df\;,$$
where H is a model, p(g|f,H) is the likelihood of *f* in the data through the model, p(f|H) is the prior knowledge about the unknown quantity *f*, and p(f|g,H) called the posterior is the result of the combination of the likelihood and the prior. The denominator p(g|H), called the evidence, is the overall likelihood of the model in the data *g*.

When there are some hyperparameters, for example, the parameters of the likelihood and those of the prior law, which have also to be estimated, we have:(12)$$p\left(f,\theta |g,H\right)=\frac{p(g|f,\theta ,H)\;\;p(fb|\theta ,H)\;\;p(\theta |H)}{p(g|H)}\;\mathrm{where}\;p\left(g|H\right)=\int p\left(g|f,\theta ,H\right)\;\;p\left(f|\theta ,H\right)\;d\theta \;df$$

This is called the joint posterior law of all the unknowns. From that joint posterior distribution, we may also obtain the marginals:(13)p(f|g,H)=∫∫p(f,θ|g,H)dfandp(θ|g,H)=∫∫p(f,θ|g,H)df.

### 7.2. Gaussian Priors Case

To be more specific, let us consider the case of linear inverse problems g=Hf+ϵ.

Then, assuming Gaussian noise, we have: (14)$$p\left(g|f\right)=N\left(g|Hf,v_{\epsilon }I\right)\propto \exp\left\{\frac{-1}{2v_{\epsilon }}{∥g-Hf∥}_{2}^{2}\right\}\;.$$

Assuming also a Gaussian prior: (15)$$p\left(f\right)\propto \exp\left\{\frac{-1}{2v_{f}}{∥f∥}_{2}^{2}\right\}\;\mathrm{or}\;\exp\left\{\frac{-1}{2v_{f}}{∥Df∥}_{2}^{2}\right\}\;,$$
it is easy to see that the posterior is also Gaussian, and the MAP and posterior mean (PM) estimates become the same and can be computed as the minimizer of: $J\left(f\right)={∥g-Hf∥}_{2}^{2}+\lambda R\left(f\right)$:(16)$$\;p\left(f|g\right)\propto \exp\left\{\frac{-1}{2v_{\epsilon }}J\left(f\right)\right\}\to {\hat{f}}_{MAP}=\arg\max_{f}\left\{p(f|g)\right\}=\arg\min_{f}\left\{J(f)\right\}\;.$$

In summary, we have: (17)$$\left\{\begin{array}{c} p\left(g|f\right)=N\left(g|Hf,v_{\epsilon }I\right)\\ p\left(f\right)=N\left(f|0,v_{f}I\right) \end{array}\right.⟶\left\{\begin{array}{c} p\left(f|g\right)=N\left(f|\hat{f},\hat{\Sigma }\right)\\ \hat{f}={\left[H^{t}H+\lambda I\right]}^{-1}H^{t}g\\ \hat{\Sigma }=v_{\epsilon }{\left[H^{t}H+\lambda I\right]}^{-1},\;\lambda =\frac{v_{\epsilon }}{v_{f}} \end{array}\right.\;.$$

This case is also summarized in (Figure 6).

We may note that, in this case, we have an analytical expression for the posterior law, which is also a Gaussian law and entirely specified by its mean $\hat{f}$ and covariance $\hat{\Sigma }$. However, for great dimensional problems where *f* is a great size vector, the computation of $\hat{\Sigma }$ can become very costly. The computation of the posterior mean $\hat{f}$ can be done by optimization as it is the same as the MAP solution.

### 7.3. Gaussian Priors with Unknown Parameters

For the case where the hyperparameters $v_{\epsilon }$ and $v_{f}$ are unknown (unsupervised case), we can derive the following: (18)$$\left\{\begin{array}{c} p\left(g|f,v_{\epsilon }\right)=N\left(g|Hf,v_{\epsilon }I\right)\\ p\left(f|v_{f}\right)=N\left(fb|0,v_{f}I\right)\\ p\left(v_{\epsilon }\right)=IG\left(v_{f}|\alpha_{\epsilon_{0}},\beta_{\epsilon_{0}}\right)\\ p\left(v_{f}\right)=IG\left(v_{f}|\alpha_{f_{0}},\beta_{f_{0}}\right) \end{array}\right.⟶\left\{\begin{array}{c} p\left(f|g,v_{\epsilon },v_{f}\right)=N\left(f|\hat{f},\hat{\Sigma }\right)\\ \hat{f}={\left[H^{t}H+\hat{\lambda }I\right]}^{-1}H^{t}g\\ \hat{\Sigma }=v_{\epsilon }h{\left[H^{t}H+\hat{\lambda }I\right]}^{-1},\;\hat{\lambda }=\frac{v_{\epsilon }h}{v_{f}h}\\ p\left(v_{\epsilon }|g,f\right)=IG\left(v_{\epsilon }|{\tilde{\alpha }}_{\epsilon },{\tilde{\beta }}_{\epsilon }\right)\\ p\left(v_{f}|g,f\right)=IG\left(v_{f}|{\tilde{\alpha }}_{f},{\tilde{\beta }}_{f}\right)\\ {\tilde{\alpha }}_{\epsilon },{\tilde{\beta }}_{\epsilon },{\tilde{\alpha }}_{f},{\tilde{\beta }}_{f} \end{array}\right.\;,$$
where all the details and, in particular, the expressions for ${\tilde{\alpha }}_{\epsilon },{\tilde{\beta }}_{\epsilon },{\tilde{\alpha }}_{f},{\tilde{\beta }}_{f}$ can be found in [19]. As we can see, the expressions of $\hat{f}$ and $\hat{\Sigma }$ are the same as in previous case, except that the values of $v_{\epsilon }h$, $v_{f}h$, and $\hat{\lambda }$ have to be updated. They are obtained from the conditionals $p(v_{\epsilon }|g,f)$ and $p(v_{f}|g,f)$ at each iteration.

This case is also summarized in Figure 7.

The joint posterior can be written as:(19)$$p\left(f,v_{\epsilon },v_{\xi }|g\right)\propto \exp\left\{-J(f,v_{\epsilon },v_{\xi })\right\}\;.$$

From this expression, we have different possible expansions:JMAP: alternate optimization with respect to $f,v_{\epsilon },v_{f}$:
(20)$$J\left(f,v_{\epsilon },v_{f}\right)=\frac{1}{2v_{\epsilon }}{∥g-Hf∥}_{2}^{2}+\frac{1}{2v_{f}}{∥f∥}_{2}^{2}+\left(\alpha_{\epsilon_{0}}+1\right)\ln v_{\epsilon }+\frac{\beta_{\epsilon_{0}}}{v_{\epsilon }}+\left(\alpha_{f_{0}}+1\right)\ln v_{f}+\frac{\beta_{f_{0}}}{v_{f}}\;.$$Each iteration can be done in two steps:1.Fix $v_{\epsilon }$ and $v_{f}$ to previous values and optimize with respect to *f*;2.Fix *f* and optimize with respect to $v_{\epsilon }$ and $v_{f}$.The first step results in a quadratic criterion with respect to *f*, which results in an analytical expression for the solution, which can be used for small-dimension problems, or it can be optimized easily by any gradient-based algorithm. The second step, in this case, results in two separate explicit solutions: one for $v_{\epsilon }$ and one for $v_{f}$.Gibbs sampling and Markov Chain Monte Carlo (MCMC):
(21)$$f∼p(f,v_{\epsilon },v_{f}|gb\to v_{\epsilon }∼p\left(v_{\epsilon }|g,f\right)\to v_{f}∼p\left(v_{f}|g,f\right)\;.$$These steps can be done using the expressions of the conditional given in Equation (Equation 18). These methods are used generally when we not only want to have a point estimator such as MAP or the posterior mean but also to quantify the uncertainties by estimations of the variances and covariances.Variational Bayesian Approximation (VBA): approximate $p(f,v_{\epsilon },v_{f}|g)$ by a separable one $q\left(f,v_{\epsilon },v_{f}\right)=q_{1}\left(f|v_{\epsilon },v_{f}\right)q_{2}\left(v_{\epsilon }\right)q_{3}\left(v_{f}\right)$ minimizing KL(q|p) [19,55,56,57,58]. We can see that the alternate optimization of $KL(q_{1},q_{2},q_{3}|p)$ with respect to $q_{1}$, $q_{2}$, and $q_{3}$ result in the same expressions as in Equation (Equation 18), only the expressions for updating the parameters ${\tilde{\alpha }}_{\epsilon },{\tilde{\beta }}_{\epsilon },{\tilde{\alpha }}_{f}$, and ${\tilde{\beta }}_{f}$ are different.The Approximate Bayesian Computation (ABC) method and, in particular, the VBA and mean-field-approximation methods are used when Gibbs sampling and MCMC methods are too expensive and we still want to quantify uncertainties, for example, estimating the variances.

## 8. Imaging inside the Body: From Data Acquisition to Decision

To introduce the link between the different model-based methods and the machine-learning tools, let us consider the case of medical imaging, from the acquisition to the decision steps:Data acquisition:Object f→CTscan,MRI,TEP,US,Microwaveimaging→Date gImage reconstruction by analytical methods:
Data g→Reconstruction→Image f^Post-processing (segmentation, contour detection, and selection of region of interest):
Imagef^→Segmentation→z^Understanding and decision:
Imagef^Segmentation z^→InterpretationDecision→Tumor orNot Tumor

### 8.1. Bayesian Joint Reconstruction and Segmentation

The questions now are: can we join any of these steps? Can we go directly from the image to the decision? For the first one, the Bayesian approach can provide a solution:
Data g→ReconstructionSegmentation →Reconstruction f^→Segmentation z^

The main tool here is to introduce a hidden variable that can represent the segmentation. A solution is to introduce a classification hidden variable *z* with $z_{j}=\left\{1,2,\cdots ,K\right\}$, which can be used to show the segmented image. See Figure 8.

Figure 8 and Figure 9 summarize this scheme.

A few comments for these relations:p(g|f,z) does not depend on *z*, so it can be written as p(g|f).We used a Markovian Potts model for p(z) to obtain more compact homogeneous regions [18,19].If we choose for p(f|z) a Gaussian law, then p(f,z|g) becomes a Gauss–Markov–Potts model [19].We can use the joint posterior p(f,z|g) to infer on (f,z): we may just do JMAP:$\left(\hat{f},\hat{z}\right)=\arg\max_{}\left\{p(f,z|g)\right\}$ or trying to access to the expected posterior values by using the Variational Bayesian Approximation (VBA) techniques [17,19,58,59,60,61,62].When the iterations finished, we obtain an estimate of the reconstructed image *f* and its segmentation *z* when using JMAP and also the covariance of *f* as well as the parameters of the posterior laws of *z*

This scheme can be extended to consider the estimation of the hyperparameters too. Figure 10 shows this.

Again, here, we can use the joint posterior p(f,z,θ|g) to infer on all the unknowns. When the iterations are finished, we get an estimate of the reconstructed image *f*, its segmentation *z*, as well as all the unknown parameters such as the means and variances of the reconstructed image at each of its segments. Giving more details is out of the focus of this overview article. They can be found more specifically in [17].

### 8.2. Advantages of the Bayesian Framework

Between the main advantages of the Bayesian framework for inverse problems, we can mention the following:Large flexibility of prior models.-Smoothness (Gaussian and Gauss–Markov).-Direct sparsity (double exp., heavy-tailed distributions).-Sparsity in the transform domain (double exp., heavy-tailed distributions on the WT coefficients).-Piecewise continuous (DE or Student-t on the gradient).-Objects composed of only a few materials (Gauss–Markov–Potts), ...Possibility of estimating hyperparameters via JMAP or VBA.Natural ways to take account for uncertainties and to quantify the remaining uncertainties.

### 8.3. Imaging inside the Body: From Data to Decision: Classical or Machine Learning

From previous sections, we see that we have many solutions to go from data to an image by inversion (image reconstruction), then extraction of interesting features (segmentation), and finally the interpretation and decision. The question that we may ask now is: *can we do all together in a more easy way?* Machine-learning and artificial-intelligence tools may propose such a solution. See Figure 11. To be able to use ML to go from data to decision, there is a crucial need of a great and rich database obtained by experts to let the machine learn from that great database. In the next section, we add a little more in detail to see the advantages, limitations, and drawbacks.

## 9. Machine Learning’s Basic Idea

The main idea in machine learning is first to learn from a great number of data-decisions: $(g_{i},d_{i}),i=1,\cdots N$: Learning date(gi,di)i=1N→Learning stepThe weights W of the NN are obtained→W
and then, when a new case (Test $g_{j}$) appears, it uses the learned weights *W* to give a decision $d_{j}$
Test case dategj→Testing stepThe learned weights W are used→Tumor orNot Tumordj

Figure 12 shows the main process of ML.

Nowadays, ML methods and tools have made great progress in many different areas of applications. There is no need here to go more in detail, instead mentioning a few main components of all of them. Between the basic tasks we can mention:Classification (supervised and semi-supervised);Clustering (unsupervised classification when the data do not yet have labels);Regression (continuous parameter estimation).

Figure 13 shows these three main tasks.

Between the existing ML tools, we may mention: support vector machines (SVM), decision-tree learning (DT), artificial neural networks (ANN), Bayesian networks (BN), HMM and random forest (RF), mixture models (GMM, SMM, etc.), KNN, Kmeans, etc.

Additionally, the combination of Imaging technology and systems, image processing, computer vision, machine learning, and artificial intelligence has been the seed for much great progress in all areas of health and our environment. The frontiers between science and technology has become less precise as is shown in Figure 14.

Between the machine-learning tools using NN, the convolutional NN (CNN), recurrent NN (RNN), deep learning (DL), generative artificial networks (GAN) had greater success in different area such as speech recognition, computer vision, and specifically in segmentation, classification and clustering, and in multi-modality and cross-domain information fusion.

However, there are still many limitations: a lack of interpretability, reliability, and uncertainty and no reasoning and explaining capabilities. To overcome this, there os still much to do with the fundamentals.

## 10. Neural Networks, Machine Learning, and Inverse Problems

### 10.1. Neural Networks

Let us start this section with a few words on neurons and neural networks. The following figures show the basic idea. The following figure shows the main idea about a neuron in a mathematical framework. Figure 15 shows this graphically.

Figure 16 shows the components of a neuron and an example of a two-layer NN.

### 10.2. NN and Learning

A neural network can be used for modeling a universal relation between its inputs *X* and outputs *Y*. This model can be written as $Y=F_{W}\left(X\right)$ where *W* represents the parameters of the model represented by the weights of the network nodes relation. They are commonly used for:Classification (supervised learning)A set of data $\left\{\left(x_{i},y_{i}\right)\right\}$ with labels (classes) $\left\{c_{i}\right\}$ are given. The objective during the training is to use them for training the network, which is then used for classifying a new income $\left(x_{j},y_{j}\right)$.Clustering (unsupervised learning)A set of data $\left\{\left(x_{i},y_{i}\right)\right\}$ is given. The objective is to cluster them in different classes $\left\{c_{i}\right\}$.Regression with all data (supervised learning).A set of data $\left\{\left(x_{i},y_{i}\right)\right\}$ are given. The objective is to find a function *F* describing the relation between them: F(x,y) or explicitly y=F(x) for any *x* (extrapolation or interpolation).

### 10.3. Modeling, Identification, and Inversion

Here, we make a connection between the classical and ML tools and show the links between forward modeling and inversion or inference, model identification and learning or training, and inversion and using the NN:Forward modeling and inversion
f→Forwardmodeling→datag∥datag→Inversioninference→f^Identification of a system and the training step of NN
f→IdentificationW→datag∥{gi,fi}→Learningtraining→Learnedmodel WInversion (inference) or using the NN-trained model
g→InversionInference→f^∥g⟶LearnedModel W⟶f^

## 11. ML for Inverse Problems

To show the possibilities of the interaction between inverse problems’ methods, machine learning, and NN methods, the best way is to give a few examples.

### 11.1. First Example: A Linear One- or Two-Layer Feed-Forward NN

The first one is the case of linear inverse problems and quadratic regularization of the Bayesian with Gaussian priors. The solution has an analytic expression, and we have the following relations: $$g=Hf+\epsilon ⟶\hat{f}={\left(H^{t}H+\lambda I\right)}^{-1}H^{t}g=A\;g=BH^{t}\;g\;\;\mathrm{or}\;\mathrm{still}\;\;\hat{f}=H^{t}{\left(\frac{1}{\lambda }HH^{t}+I\right)}^{-1}g=H^{t}C\;g$$
where $A={\left(H^{t}H+\lambda I\right)}^{-1}H^{t}$, $B={\left(H^{t}H+\lambda I\right)}^{-1}$ and $C={\left(\frac{1}{\lambda }HH^{t}+I\right)}^{-1}$.

These relations can be presented schematically as
g→A→f^,g→Ht→B→f^,g→C→Ht→f^

As we can see, these relations directly induce a linear feed-forward NN structure. In particular, if *H* represents a convolution operator, then $H^{t}$, $H^{t}H$, and $HH^{t}$ are too, as well as the operators *B* and *C*. Thus, the whole inversion can be modeled by CNN [63].

For the case of computed tomography (CT), the first operation is equivalent to an analytic inversio;, the second corresponds to back-projection first followed by 2D filtering in the image domain; and the third corresponds to to the famous filtered back-projection (FBP), which is implemented on classical CT scans. These three cases are illustrated on Figure 17.

### 11.2. Second Example: Image Denoising with a Two-Layer CNN

The second example is the denoising g=f+ϵ with an L1 regularizer: (22)$$\hat{f}=D\hat{z}\;and\;\hat{z}=\arg\min_{z}\left\{J(z)\right\}\;with\;J\left(z\right)={∥g-Dz|+\lambda ∥z∥}_{1}\;,$$
where ***D*** is a filter, i.e., a convolution operator. This can also be considered as the MAP estimator with a double exponential prior. It is easy to show that the solution can be obtained by a convolution followed by a thresholding [64,65,66].
$$\hat{f}=D\hat{z}\;\mathrm{and}\;\hat{z}=S_{\frac{1}{\lambda }}\left(D^{t}g\right)$$
where $S_{\lambda }$ is a thresholding operator.
g→Dt→Thresholding→z^→D→f^or equivalentlyg→Two-layer CNN→f^

### 11.3. Third Example: A Deep-Learning Equivalence

One of the classical iterative methods in linear inverse-problems algorithms is based on a gradient-based method to optimize $J\left(f\right)={∥g-Hf∥}^{2}$: (23)$$f^{\left(k+1\right)}=f^{\left(k\right)}+\alpha H^{t}\left(g-Hf^{\left(k\right)}\right)=\alpha H^{t}g+\left(I-\alpha H^{t}H\right)f^{\left(k\right)}\;,$$
where the solution of the problem is obtained recursively. Everybody knows that when the forward model operator ***H*** is singular or ill-conditioned, this iterative algorithm starts by converging, but it may diverge easily. One of the experimental methods to obtain an acceptable approximate solution is just to stop the iterations after *K* iterations. This idea can be translated to a deep-learning NN by using *K* layers. Each layer represents one iteration of the algorithm. See Figure 18 and Figure 19.

This DL structure can easily be extended to a regularized criterion: $J\left(f\right)=\frac{1}{2}{∥g-Hf∥}^{2}+\lambda {∥Df∥}^{2}$, where
(24)$$f^{\left(k+1\right)}=f^{\left(k\right)}+\alpha \left[H^{t}\left(g-Hf^{\left(k\right)}\right)-\lambda D^{t}D\right]=\alpha H^{t}g+\left(I-\alpha H^{t}H-\alpha \lambda D^{t}D\right)f^{\left(k\right)}\;.$$We just need to replace $\left(I-\alpha H^{t}H\right)$ by $\left(I-\alpha H^{t}H-\alpha \lambda D^{t}D\right)$.

This structure can also be extended to all the sparsity-enforcing regularization terms such as $\ell_{1}$ and total variation (TV) using appropriate algorithms such as ISTA, FISTA, ADMM, etc. by replacing the update expression and by adding a NL operation much like the ordinary NNs. A simple example is given in the following subsection.

### 11.4. Fourth Example: $\ell_{1}$ Regularization and NN

Let us consider the linear inverse problem g=Hf+ϵ with $\ell_{1}$ regularization criterion: $$J\left(f\right)={∥g-Hf∥}_{2}^{2}+\lambda {∥f∥}_{1}\;,$$
and an iterative optimization algorithm, such as ISTA: $$f^{\left(k+1\right)}=Prox_{\ell_{1}}\left(f^{\left(k\right)},\lambda \right)\;\overset{▵}{=}\;S_{\lambda \alpha }\left(\alpha H^{t}g+\left(I-\alpha H^{t}H\right)f^{\left(k\right)}\right)\;,$$
where $S_{\theta }$ is a soft thresholding operator, and $\alpha \leq |eig(H^{t}H)|$ is the Lipschitz constant of the normal operator. When *H* is a convolution operator, then:$\left(I-\alpha H^{t}H\right)f^{\left(k\right)}$ can also be approximated by a convolution and thus considered as a filtering operator;$\frac{1}{\alpha }H^{t}g$ can be considered as a bias term and is also a convolution operator; and$S_{\theta =\lambda \alpha }$ is a nonlinear point-wise operator. In particular, when *f* is a positive quantity, this soft thresholding operator can be compared to the ReLU activation function of NN. See Figure 20.

In all these three examples, we directly could obtain the structure of the NN from the forward model and known parameters. However, in these approaches, there are some difficulties, which consist in the determination of the structure of the NN. For example, in the first example, obtaining the structure of ***B*** depends on the regularization parameter λ. The same difficulty arises for determining the shape and the threshold level of the thresholding bloc of the network in the second example. The same need of the regularization parameter as well as many other hyperparameters is necessary to create the NN structure and weights. In practice, we can decide, for example, on the number and structure of a DL network, but as their corresponding weights depend on many unknown or difficult to fix parameters, ML may become of help. In the following, we first consider the training part of a general ML method. Then, we see how to include the physics-based knowledge of the forward model in the structure of learning.

## 12. ML General Approach

The ML approach can become helpful if we could have a great amount of data: inputs–outputs $\left\{{\left(f,g\right)}_{k},k=1,2,\ldots ,K\right\}$ examples. Thus, during the training step, we can learn the coefficients of the NN and then use it for obtaining a new solution $\hat{f}$ for a new data *g*.

The main issue is the number of data input–output examples $\left\{{\left(f,g\right)}_{k},k=1,2,\ldots ,K\right\}$ we can have for the training step of the network.

### Fully Learned Method

Let us consider a one-layer NN where the relation between its input $g_{k}$ and output $f_{k}$ is given by $f_{k}=\phi \left(Wg_{k}\right)$ where *W* is the weighting parameters of the NN, and ϕ is the point-wise non-linearity function of the output NN output layer. The estimation of *W* from the training data in the learning step is done by an optimization algorithm, which optimizes a loss function L defined as
(25)$$L=\sum_{k=1}^{K}\ell_{k}\left(f_{k},\phi \left(Wg_{k}\right)\right)$$
with
(26)$$\ell_{k}(f_{k},\phi \left(Wg_{k}\right)={∥f_{k}-\phi \left(Wg_{k}\right)∥}^{2}\;,$$
a quadratic distance or any other appropriate distance or divergence or a probabilistic one
(27)$$\ell_{k}(f_{k},\phi \left(Wg_{k}\right)=E\left\{∥f_{k},\phi \left(Wg_{k}\right){∥}^{2}\right\}\;.$$

When the NN is trained and we obtain the weights $\hat{W}$, then we can use it easily when a new case (Test $g_{j}$) appears, just by applying: $f_{k}=\phi \left(\hat{W}g_{k}\right)$. These two steps of training and using (also called testing) are illustrated in Figure 21.

The scheme that we presented is general and can be extended to any multi-layer NN and DL. In fact, if we had a great number of data-ground-truth examples $\left\{{\left(f,g\right)}_{k},k=1,2,\ldots ,K\right\}$ with *K* much more than the number of elements $W_{m,n}$ of the weighting parameters *W*, then, we would not even have any need for forward model ***H***. This can be possible for very low dimensional problems [67,68,69,70]. However, in general, in practice, we do not have enough data. So, some prior or regularizer is needed to obtain a usable solution. This can be done just by adding a regularizer R(W) to the loss function (Equation 25) and (Equation 26), but we can also use the physics of the forward operator ***H***.

## 13. Physics-Based ML

As mentioned above, in general, in practice, a rich-enough and complete data set is not often available, particularly for inverse problems. So, some prior or regularizer is needed to obtain a usable solution. Using a regularizer R(W) to the loss function (Equation 25) is good but is not enough. We have to use the physics of the forward operator *H*. This can be done in different ways.

### 13.1. Decomposition of the NN Structure to Fixed and Trainable Parts

The first, easiest, and understandable method consists in decomposing the structure of the network *W* in two parts: a fixed part and a learnable part. As the simplest example, we can consider the case of the analytical expression of the quadratic regularization: $\hat{f}={\left(HH^{t}+\lambda DD^{t}\right)}^{-1}H^{t}g=BH^{t}g$ which suggests to have a two-layer network with a fixed part structure $H^{t}$ and a trainable one $B={\left(HH^{t}+\lambda DD^{t}\right)}^{-1}$. See Figure 22.

It is interesting to note that, in X-ray computed tomography (CT), the forward operator ***H*** is called *projection*, the adjoint operator $H^{t}$ is called *back-projection (BP)*, and the ***B*** operator is assimilated to a 2D filtering (convolution).

### 13.2. Using Singular-Value Decomposition of Forward and Backward Operators

Using the eigenvalues and eigenvectors of the pseudo or generalized inverse operators
(28)$$H^{†}={\left[H^{t}H\right]}^{-1}H^{t}\;\;\mathrm{or}\;\;H^{†}=H^{t}{\left[HH^{t}\right]}^{-1}$$
and singular-value decomposition (SVD) of the operators $\left[H^{t}H\right]$ and $\left[HH^{t}\right]$ give another possible decomposition of the NN structure. Let us note
(29)$$HH^{t}=U\Delta V^{\prime }\;\mathrm{or}\;\mathrm{equivalently}\;H^{t}H=V\Delta U^{\prime }\;,$$
where Δ is a diagonal matrix containing the singular values, *U* and *V*, containing the corresponding eigenvectors. This can be used to decompose the *W* to four operators: (30)$$W=V^{\prime }\Delta UH^{t}\;\;\mathrm{or}\;\;W=H^{t}V\Delta U^{\prime }\;,$$
where three of them can be fixed, and only one Δ can be trainable. It is interesting to know that when the forward operator *H* has a shift-invariant (convolution) property, then the operators *U* and $V^{\prime }$ will correspond to the FT and IFT operators, respectively, and the diagonal elements of Λ correspond to the FT of the impulse response of the convolution forward operator. So, we will have a fixed layer corresponding to $H^{t}$, which can be interpreted as a matched filtering, and a fixed FT layer, which is a feed-forward linear network, a trainable filtering part corresponding to the diagonal elements of Λ, and a forth fixed layer corresponding to IFT. See Figure 23.

#### DL Structure Based on Iterative Inversion Algorithm

Using the iterative gradient-based algorithm with a fixed number of iterations for computing a GI or a regularized one as explained in previous section can be used to propose a DL structure with *K* layers, *K* being the number of iterations before stopping. Figure 24 shows this structure for a quadratic regularization, which results to a linear NN and Figure 25 for the case of $\ell_{1}$ regularization.

## 14. Conclusions and Challenges

Signal and image processing (SIP), imaging systems (IS), computer vision (CV), machine learning (ML), and artificial intelligence (AI) have made great progress in the last forty years. The first category of the methods in signal and image processing was based on linear transformation followed by a thresholding or windowing and coming back. The second generation was model based: the forward-modeling and the inverse-problems approach. The main successful approach was based on regularization methods using a combined criterion. The third generation was model based but probabilistic and used the Bayes rule, which is the Bayesian approach.

Classical methods for inverse problems are mainly based on regularization methods, particularly those that are based on the optimization of a criterion with two parts: a data-model matching part and a regularization term. A great number of methods have been proposed for choosing these two parts and proposing appropriate optimization algorithms. A Bayesian Maximum A Posteriori (MAP) interpretation for these regularization methods can be given where these two terms correspond to the likelihood and prior probability models, respectively.

The Bayesian approach gives more flexibility in different aspects: (i) in choosing these terms and, in particular, the prior term via hierarchical models and hidden variables; (ii) a more-extended class of prior models can be obtained, particularly via the hierarchical prior models; (iii) determination of the regularization parameter, and more generally all the hyperparameters, can also be estimated; (iv) all the uncertainties are accounted for, and all the remaining uncertainties can be evaluated.

However, the Bayesian computations can become very heavy computationally, particularly when we want compute the uncertainties (variances and covariances) and when we want also to estimate the hyperparameters. Recently, the machine-learning (ML) methods have become a good help for some aspects of these difficulties.

Nowadays, ML, neural networks (NN), convolutional NN (CNN), and deep-learning (DL) methods have obtained great success in classification, clustering, object detection, speech and face recognition, etc. However, they need a great number of training data, still lack explanation, and they may fail very easily.

For inverse problems, they still need progress. In fact, using only data-based NN without any specific structure coming from the forward model (physics) is just an illusion. However, the progress arrives via their interaction with the model-based methods. In fact, the success of CNN and DL methods greatly depends on the appropriate choice of the network structure. This choice can be guided by the model-based methods.

In this work, a few examples of such interactions are described. As we could see, the main contribution of ML and NN tools can be in reducing the costs of the inversion method when an appropriate model is trained. However, to obtain a good model, there is a need for sufficiently rich data and a good network structure obtained from the physics knowledge of the problem in hand.

For inverse problems, when the forward models are non-linear and complex, NN and DL may be of great help. However, we may still need to choose the structure of the NN via an approximate forward model and approximate Bayesian inversion. 

## Figures and Tables

**Figure 1 entropy-23-01673-f001:**
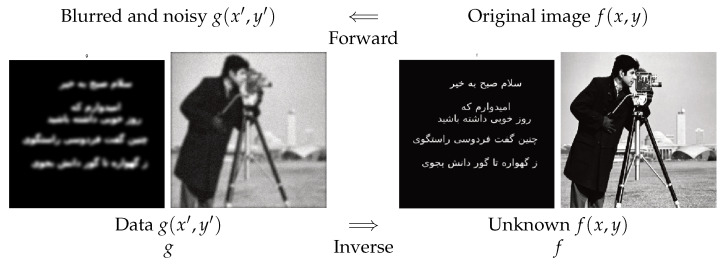
Forward and inverse problems in image restoration. Forward operation is a *convolution*, and the inverse operation is called *deconvolution*.

**Figure 2 entropy-23-01673-f002:**
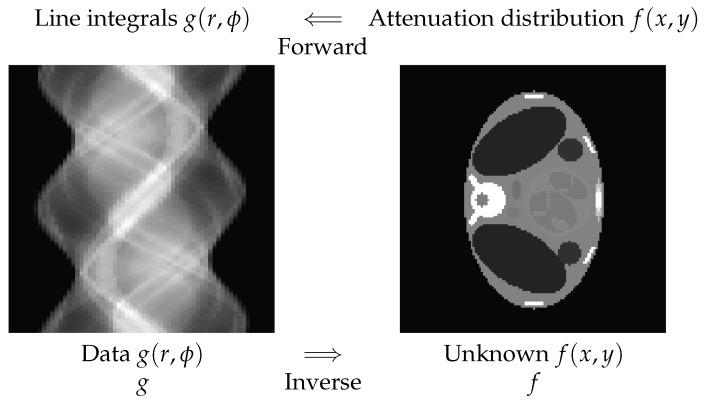
Forward and inverse problems in computed tomography. The horizontal axis on the left is *r*, the vertical is ϕ, and the values of g(r,ϕ) are presented as the gray levels. On the right, the object section f(x,y) is presented. Forward operation is called *projection*, and the inversion process is called *image reconstruction*.

**Figure 3 entropy-23-01673-f003:**
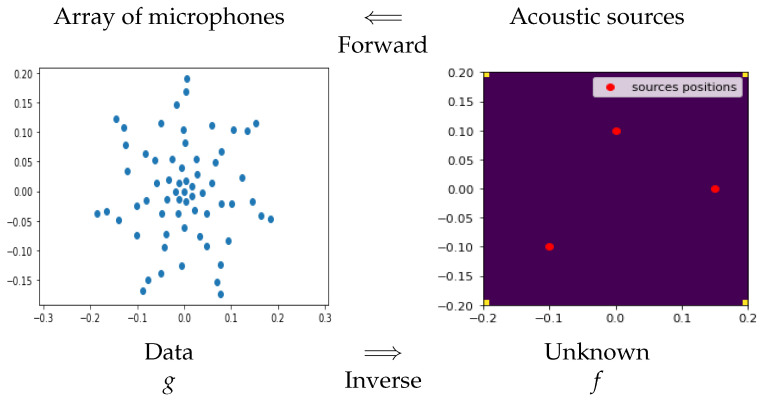
Forward and inverse problems in acoustical imaging. Each microphone receives the sum of the delayed sources’ sounds. The inverse problem is to estimate the sources’ distribution from the received signals by the microphones array.

**Figure 4 entropy-23-01673-f004:**
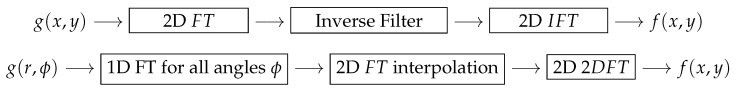
Transform-based analytical methods. Two examples are given: image deconvolution by inverse filtering and image reconstruction in CT by using the relation between RT and FT.

**Figure 5 entropy-23-01673-f005:**
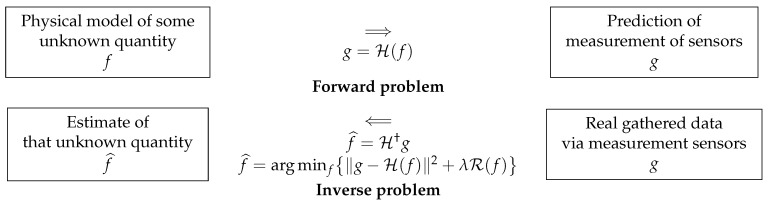
Model-based methods: forward and inverse problems. The solution of the inverse problem is defined either by the generalized inversion or by a regularization method.

**Figure 6 entropy-23-01673-f006:**
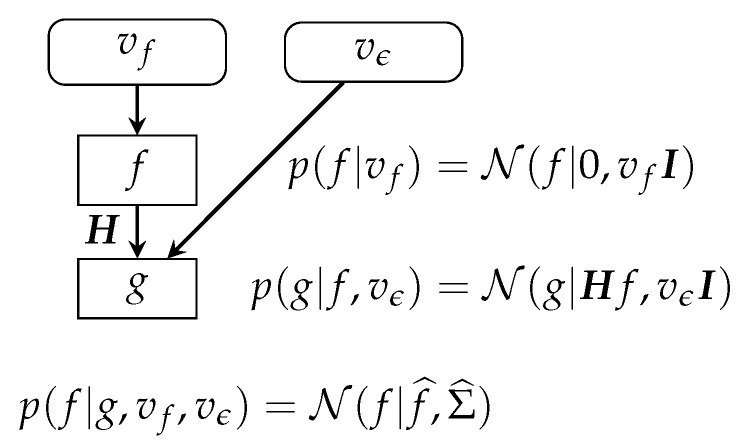
Bayesian-inference scheme in linear systems and Gaussian priors. The posterior is also Gaussian, and all the computations can be done analytically.

**Figure 7 entropy-23-01673-f007:**
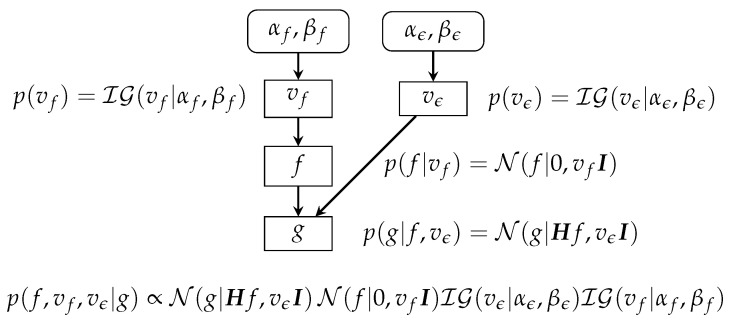
Bayesian-inference scheme in linear systems and Gaussian priors. The posterior is also Gaussian, and all the computations can be done analytically.

**Figure 8 entropy-23-01673-f008:**
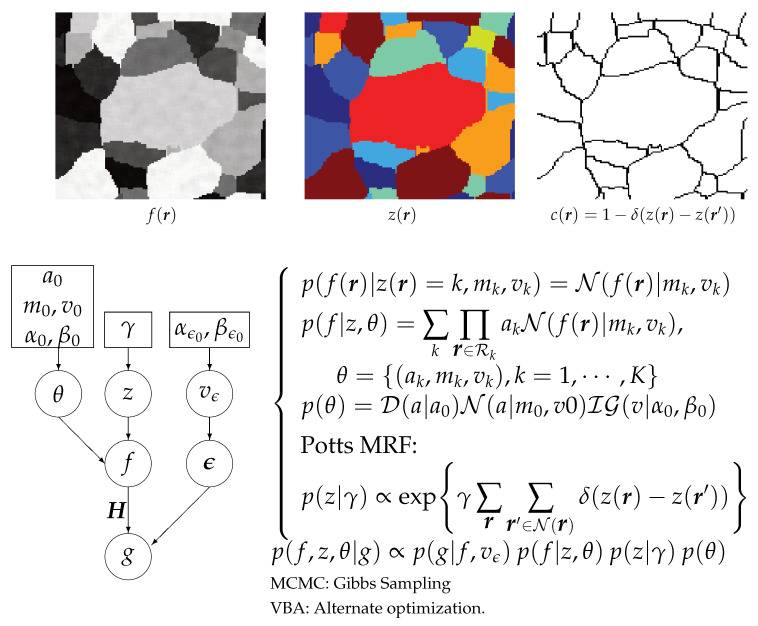
Gauss–Markov–Potts prior model for Bayesian image reconstruction and segmentation.

**Figure 9 entropy-23-01673-f009:**
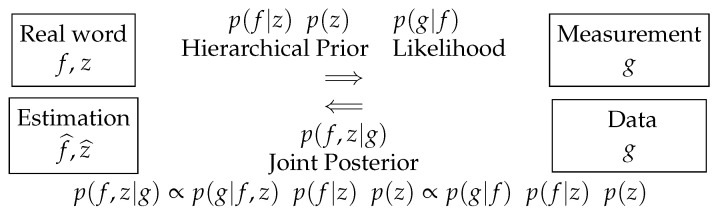
Bayesian approach with hierarchical prior model for joint reconstruction and segmentation.

**Figure 10 entropy-23-01673-f010:**
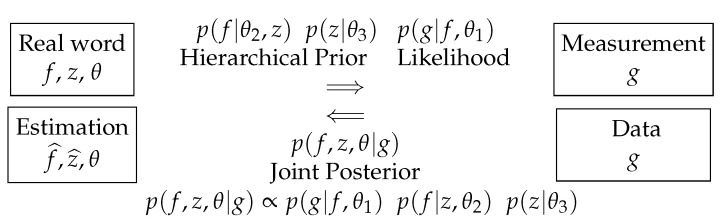
Advanced Bayesian approach for joint reconstruction, segmentation and parameter estimation.

**Figure 11 entropy-23-01673-f011:**
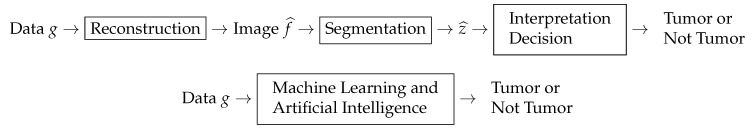
Two approaches going from the data to decision. Top: from data, first, reconstruct an image via inversion, then post-process to obtain segmentation. Perform pattern recognition to extract the contours of the region of interest and finally make a decision. Bottom: try to use machine-learning methods to go directly from data to decision.

**Figure 12 entropy-23-01673-f012:**
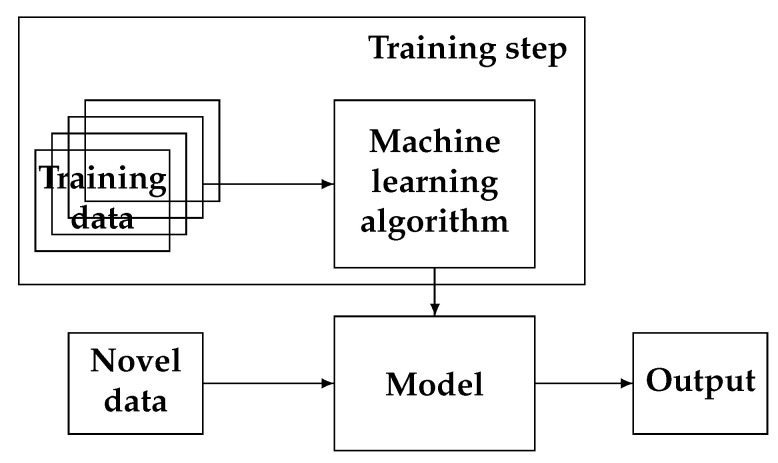
Basic machine-learning process: first, learn a model, then use it. Learning step needs a rich-enough database, which is expensive. When the model is learned and tested, its use is easy and fast, and its cost is low.

**Figure 13 entropy-23-01673-f013:**
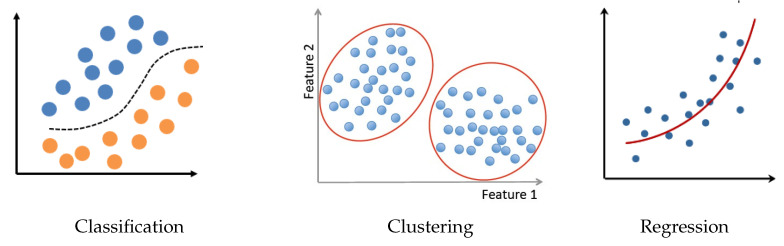
Basic machine-learning tasks: classification, clustering, and regression.

**Figure 14 entropy-23-01673-f014:**
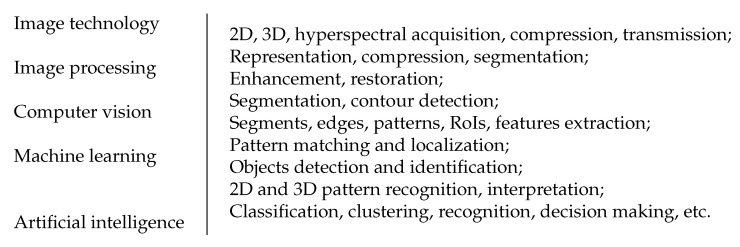
Frontiers between image technology, image processing (IP), computer vision (CV), machine Learning (ML) and artificial intelligence (AI).

**Figure 15 entropy-23-01673-f015:**
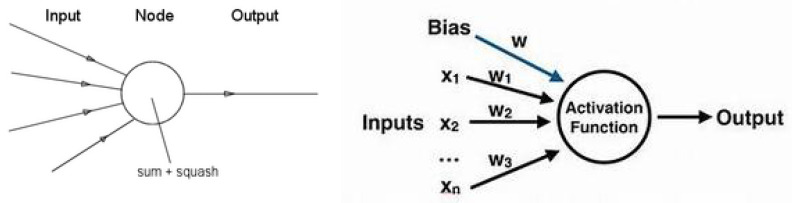
A neuron and its mathematical representation.

**Figure 16 entropy-23-01673-f016:**
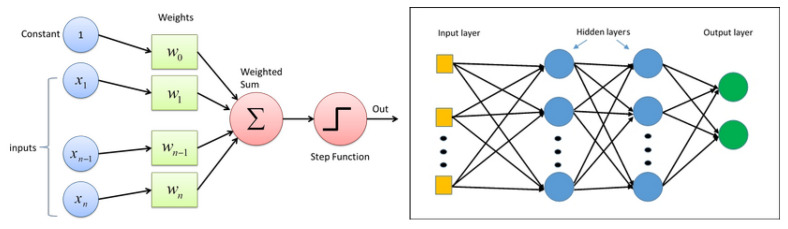
A neuron with its inputs and outputs and and a neural network with two-hidden-layer neurons.

**Figure 17 entropy-23-01673-f017:**
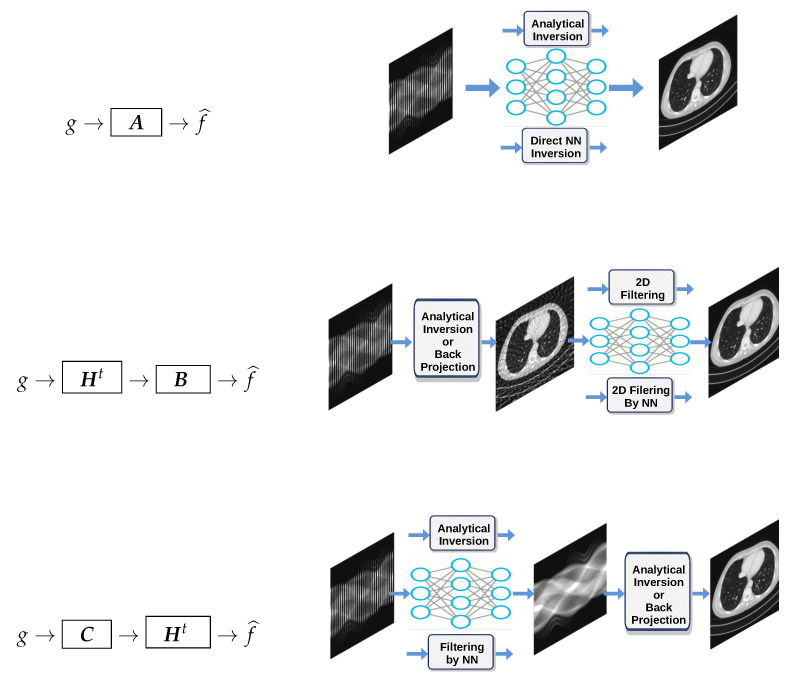
Three linear NN structures that are derived directly from quadratic regularization inversion method.

**Figure 18 entropy-23-01673-f018:**
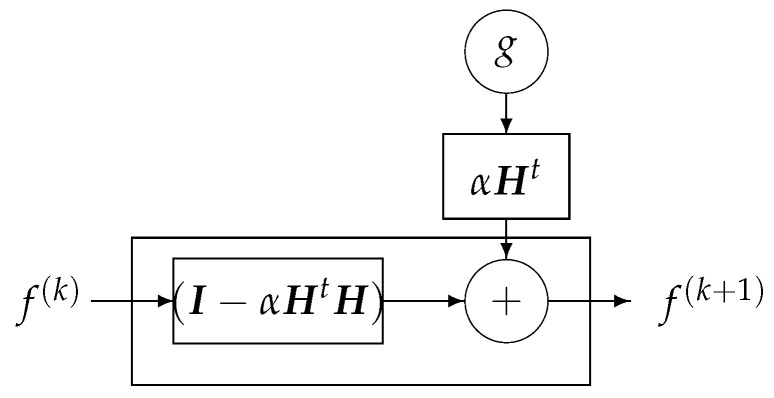
One bloc of iteration that can be considered as one layer of a NN.

**Figure 19 entropy-23-01673-f019:**
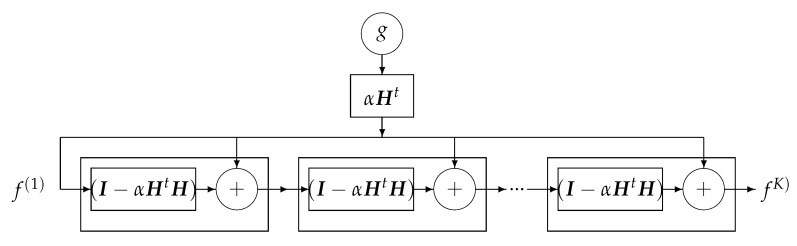
A *K* layers DL NN equivalent to *K* iterations of the basic optimization algorithm.

**Figure 20 entropy-23-01673-f020:**
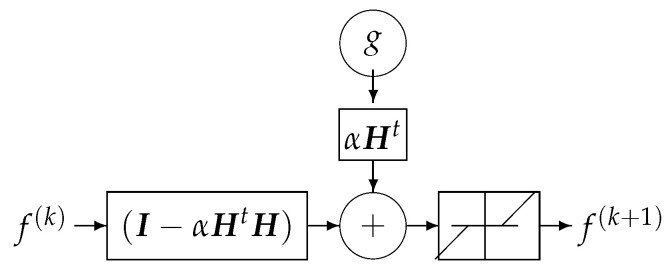
One block of a NN corresponds to one iteration of $\ell_{1}$ regularization.

**Figure 21 entropy-23-01673-f021:**
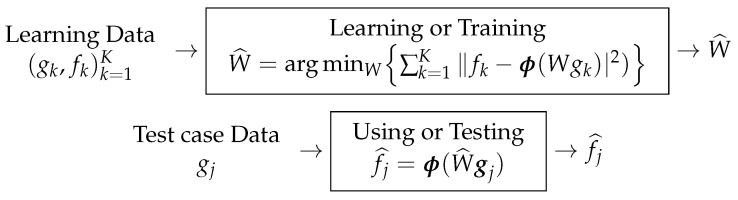
Training (**top**) and testing (**bottom**) steps in a ML approach.

**Figure 22 entropy-23-01673-f022:**
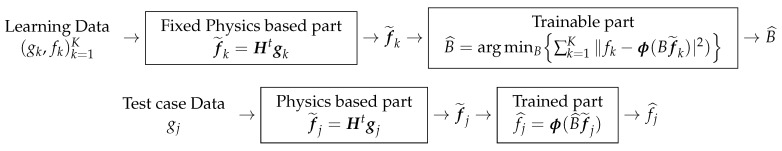
Training (**top**) and testing (**bottom**) steps in the first use of a physics-based ML approach.

**Figure 23 entropy-23-01673-f023:**
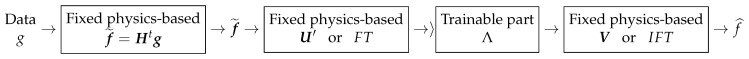
A four-layer NN with three physics-based fixed operators corresponding to $H^{t}$, $U^{\prime }$ or FT, and *V* or IFT layers and one trainable layer corresponding to Λ.

**Figure 24 entropy-23-01673-f024:**
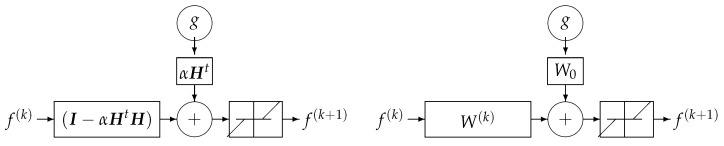
A *K* layers DL NN equivalent to *K* iterations of a basic gradient-based optimization algorithm. A quadratic regularization results in a linear NN, while a $\ell_{1}$ regularization results in a classical NN with a nonlinear activation function. Left: supervised case. Right: unsupervised case. In both cases, all the *K* layers have the same structure.

**Figure 25 entropy-23-01673-f025:**
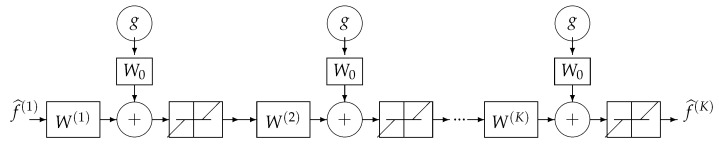
All the *K* layers of DL NN equivalent to *K* iterations of an iterative gradient-based optimization algorithm. The simplest solution is to choose $W_{0}=\alpha H$ and $W^{\left(k\right)}=W=\left(I-\alpha H^{t}H\right),\;k=1,\cdots ,K$. A more robust but more costly method is to learn all the layers $W^{\left(k\right)}=\left(I-\alpha^{\left(k\right)}H^{t}H\right),\;k=1,\cdots ,K$.

## Data Availability

Not applicable.

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
