# Peer review of "Regularization, Bayesian Inference, and Machine Learning Methods for Inverse Problems†"

_entropy, 2021, doi:10.3390/e23121673_

Round 1

Reviewer 1 Report

This work is a tutorial on inverse problems; there is no new material here. Moreover, the way is written sounds more to student notes than to a final paper or review. It makes me think this by reading a simple matrix transposition in formula (5).

The English would be improved. For instance, I don't understand the "but " in line 12 of the abstract. In general, sentences are fragmented and very few connected. The paper is also too long with many repeated sentences. For example, this sentence "the two terms of the regularization methods are distance or divergence measures, they can have a Bayesian Maximum A Posteriori (MAP) interpretation where these two terms correspond, respectively, to the likelihood and prior probability models." appears exactly the same in the abstract and on page 2.

Further many b and r appears in the symbols (starting from section 2.1 line 87. making it hard to read. Latex should be improved (see $f0$ instead of $f_0$ at page 2 line 34). Further notation is poorly defined and sometimes at all: what is $D$ in formula (9) ?

I would shorten the paper by just focusing on image reconstruction and skip all other non-relevant examples, by just indicating relevant bibliography for specific applications, for instance, on page 5 lines 113-116.

Generally, the reported methods should be illustrated by mentioning the relevant software and practical implementation.

Author Response

I agree with you that this is a tutorial paper, reflecting my tutorial presentation at MaxEnt 2021. I agree that there is no new material. For the other points, I will surely take care of eliminating the redundancies, correct the English and all the detailed valuable remarks. Here are summary of my answers to other comments:

-> Formula (5) show the generalized inverse solutions for over or under-determined cases. 

-> corrections to abstract as well as in the whole text are done. 

-> LaTex error are corrected. D is defined. 

-> As I wanted this paper as a more general review paper, I keep different examples. This also was the advice of the second reviewer. So, I expanded a little those examples.

-> The paper will then be much longer. However, I added some elements of implementation for some examples. 

Reviewer 2 Report

Overall, this paper provides a high-quality survey on the integration of machine learning and inverse problems. I enjoyed reading this paper. Please see below for my section-to-section (minor) comments.

Section 1:

1. In the introduction, the unknown f is introduced as functions. However, if divergence measures are used in the regularization term, then the unknown should be interpreted as a density function. Some discussion and clarification are needed here.

2. line 34,  it should be \Delta(f, f_0).

3. line 35, f_0 instead of f0.

Section 2:

1. In eq (2). is \delta the Dirac delta?

2. In examples 3 - 5, it will be useful to provide mathematical formulations to at least one example.

3. As a tutorial, should there be some examples to show the ill-posedness?

Section 4:

1. If I understand correctly, the coefficients of the Fourier / Radon transforms used here can have fast decay and some of the coefficients are near zero. In the inverse filter, truncation is needed to stabilize the inversion. Should this be mentioned?

Section 7:

1. In eq (9), what type of function phi is needed here? E.g., convex, monotone, etc.

2. In eq (18), it is a bit hard to follow the right half of the equation and why we need those conditionals.

Section 8:   1. In figure 8, it can be useful to split the images and equations and give more description to the equations. It is unclear what do the terms "MCMC: ..." and "VBA: ..." refer to.   2. lines 200 and 201. These two lines need to be rephrased and joined together.   Section 11:   1. lines 285 - 287, it is unclear about the meaning of this statement on convergence. Please rephrase.   2. line 289, missing period at the end of the sentence.   3. Figure 17, "One block".    

Author Response

I thank this reviewer for detailed valuable remarks, questions and corrections. I will surely take care of all of them.

Section 1:
1-> Yes, I agree. A sentence is added in page 2 to give this precision. 
2,3-> These are corrected

Section 2:
1. -> This is specified and eq. corrected as it should be a double integral. 

2. -> I added this for sources localization problem. 

3. As a tutorial, should there be some examples to show the ill-posedness?

Section 4:

1. yes, you are write. A sentence is added. 

Section 7:

1. -> Some precision is given in the sentence just after this equation. 

2. -> In this unsupervized case, the joint posterior of all the unknowns p(f,v_e,v_f|g) can be written, but the JMAP solution can only obtained by an iterative alternate optimization, where at each iteration, we may need the expressions of the conditionals given at the right hand. 
Some more explanation is added.  

Section 8:   
1. In figure 8, it can be useful to split the images and equations and give more description to the equations. It is unclear what do the terms "MCMC: ..." and "VBA: ..." refer to. 
--> Some detailed explanations are added. 

2. lines 200 and 201. These two lines need to be rephrased and joined together.   Section 11:   1. lines 285 - 287, it is unclear about the meaning of this statement on convergence. Please rephrase.   
--> Yes, it is done

2. line 289, missing period at the end of the sentence.   
3. Figure 17, "One block". 
--> Corrections are done. 

Round 2

Reviewer 1 Report

The author answered to the point that I raised.